# GaIn: Human Gait Inference for Lower Limbic Prostheses for Patients Suffering from Double Trans-Femoral Amputation

**DOI:** 10.3390/s18124146

**Published:** 2018-11-26

**Authors:** Roman Chereshnev, Attila Kertész-Farkas

**Affiliations:** Department of Data Analysis and Artificial Intelligence, Faculty of Computer Science, National Research University Higher School of Economics, 20 Myasnitskaya ulitsa, Moscow 101000, Russia; rchereshnev@hse.ru

**Keywords:** human activity recognition, gait analysis, human gait inference, wearable sensors, limb amputation, lower limbic prosthesis, machine learning, recurrent neural networks

## Abstract

Several studies have analyzed human gait data obtained from inertial gyroscope and accelerometer sensors mounted on different parts of the body. In this article, we take a step further in gait analysis and provide a methodology for predicting the movements of the legs, which can be applied in prosthesis to imitate the missing part of the leg in walking. In particular, we propose a method, called GaIn, to control non-invasive, robotic, prosthetic legs. GaIn can infer the movements of both missing shanks and feet for humans suffering from double trans-femoral amputation using biologically inspired recurrent neural networks. Predictions are performed for casual walking related activities such as walking, taking stairs, and running based on thigh movement. In our experimental tests, GaIn achieved a 4.55° prediction error for shank movements on average. However, a patient’s intention to stand up and sit down cannot be inferred from thigh movements. In fact, intention causes thigh movements while the shanks and feet remain roughly still. The GaIn system can be triggered by thigh muscle activities measured with electromyography (EMG) sensors to make robotic prosthetic legs perform standing up and sitting down actions. The GaIn system has low prediction latency and is fast and computationally inexpensive to be deployed on mobile platforms and portable devices.

## 1. Introduction

The increasing availability of wearable body sensors has led to novel scientific studies on human activity recognition and human gait analysis [1,2,3]. Human activity recognition (HAR) usually focuses on activities related to or performed by legs, such as walking, jogging, turning left or right, jumping, lying down, going up or down the stairs, sitting down, and so on. Human gait analysis (HGA), in contrast, focuses not only on the identification of activities performed by the user, but also on how the activities are performed. This is useful in exoskeleton design, sports, rehabilitation, and health care.

The walking gait cycle of a healthy human consists of two main phases: a swing phase that lasts about 38% and a stance phase that lasts about 62% of the gait cycle [4]. A good gait is related to a minimal mechanical energy consumption [5]. An unusual gait cycle can be evidence of disease; therefore, gait analysis is important in evaluating gait disorders, neurodegenerative diseases such as multiple sclerosis, cerebellar ataxia, brain tumors, etc. Multiple sclerosis patients show alterations in step size and walking speed [6]. The severity of Parkinson’s disease and stroke shows a high correlation with stride length [7]. Wearable sensors can be used to detect and measure gait-related disorders, to monitor patient’s recovery, or to improve athletic performance. For instance, EMG sensors can be used to evaluate muscle contraction force to improve performance [8,9] in running [10] and other sport fields [11]. Emergency fall events can be detected with tri-axial accelerometers attached on the waist of elderly people [12]. Accelerometers installed on the hips and legs of people with Parkinson’s disease can be used to detect freezing of gait and can prevent falling incidents [13,14,15,16].

Exoskeletons can provide augmented physical power or assistance in gait rehabilitation. In the former case, exoskeletons can be used to help firefighters and rescue workers in dangerous environments, nurses to move heavy patients [17], or soldiers to carry heavy loads [18]. Rehabilitation exoskeletons can be used to provide walking support for elderly people or can be applied in the rehabilitation of stroke or spinal cord injury [19,20]. The neuromuscular disease Cerebral Palsy, which affects the symmetry and the variability of the walking, represents the pathology that mainly needs the usage of exoskeletons/prostheses to rehabilitate the walking [21].

We introduce a new methodology, termed human gait inference (HGI), for predicting what would be the movements of amputated leg parts (thigh, shank, or foot) for causal walking-related activities such as walking, taking stairs, sitting down, standing up, etc. Limb losses occur due to (a) vascular disease (54%), including diabetes and peripheral arterial disease, (b) trauma (45%), and (c) cancer (less than 2%) [22]. Up to 55% of people with a lower extremity amputation due to diabetes will require amputation of the second leg within 2–3 years [23]. In the USA, about 2 million people live with limb loss [22].

In this article, we propose a gait inference system, called GaIn, for patients suffering at most double trans-femoral amputation. Our idea is based on the high correlation between the movements of the leg parts of people without functional gait disorder during usual activities. Figure 1 shows a nonlinear correlation between the thigh and shank angles (of the same leg) during several gait cycles measured during walking related activities. The angles of the thigh and shank are measured to the horizontal line. Consequently, it is possible to infer the movements of the lower legs (both shanks and feet) based on the movements of both thighs using machine learning methods. The GaIn system could be installed on microchip-controlled robotic leg prostheses that could be attached to patients in a non-invasive way to infer the movements of the lower limbs, as illustrated in Figure 2. Therefore, the GaIn system could help patients suffering partial or double lower limb amputation to move and walk alone.

The HGI methodology and our GaIn system are different from exoskeletons. GaIn and HGI provide methods to infer the movements of the missing lower leg parts (shanks and feet), which are directly controlled by the remaining parts of the patient’s legs (thigh). In contrast, rehabilitation exoskeletons often replay a reference gait trajectory prerecorded on healthy users, which might result in an unsuitable gait for the patient [1,19,24]. The patients’ own efforts are not taken into account. However, the exoskeletons for augmented physical strength incorporate data obtained from the whole legs of healthy users [1].

The GaIn controller consists of two components: activity recognition and gait inference. The first component recognizes whether the patient is sitting, standing, or moving. In a sitting position, GaIn does not allow any gait inference, so the legs remain motionless. However, when thigh muscle activity is detected, the controller performs a standing up activity. When the patient is standing and starts swinging one of his legs, then GaIn activates the gait inference procedure. Because the human movement is produced with neural mechanisms in the motor cortex of the human brain or spinal neural circuits [25], we believe the neurally inspired artificial neural networks could be suitable models for gait inference. Therefore, GaIn uses recurrent neural networks for inferring human gait. In addition, we designed GaIn to be fast and computationally inexpensive, performing low prediction latency. In our opinion, these features are necessary in order to be applied on mobile devices where energy consumption matters [26]. We note that turning during walking involves rotating the torso, hip, and the thighs at hip joints but not the shanks [27]; therefore, our analysis does not examine turning strategies.

The GaIn system to be efficiently used in portable systems should meet the following requirements:Low prediction latency.Smooth, continuous activity recognition within a given activity and rapid transition in between different activities.Fast and energy efficiency.

The first requirement ensures that the model is of low latency; therefore, activity prediction can be made instantly based on the latest observed data. Therefore, bidirectional models, such as bidirectional long short-term memory (LSTM) recurrent neural networks (RNN) [28] or dynamic time warping (DTW) [29] methods, are not appropriate for our aims for two main reasons: first, these bidirectional methods require a whole observed sequence before making any predictions, which would therefore increase their latency. Second, the prediction they make on a frame is based on subsequent data. Standard hidden Markov models (HMMs) have become the de facto approach for activity recognition [30,31,32,33], and they yield good performance in general. However, they do so at the expense of increased latency in prediction because Viterbi algorithms use the whole sequence, or at least some part of it, in order to estimate a series of activities (i.e., hidden states), and their time complexity is polynomial. Therefore, in our opinion, HMMs are not adequate for on-the-fly prediction because the latency of these methods can be considered rather high.

The second point is to ensure that an activity recognition method provides consistent prediction within the same activity, but changes rapidly when the activity has changed. Lester et al. [32] have pointed out that single-frame prediction methods such as decision stumps or support vector machines are prone to yielding scattered predictions. However, human activity data are time series data in nature, and subsequent data frames are highly correlated. This tremendous amount of information can be exploited simply by sequential models such as HMM and RNN, or by incorporating the sliding-window technique to single-frame methods (e.g., nearest-neighbor). In fact, the authors in [31] have pointed out that the continuous-emissions HMM-based sequential classifier (cHMM) performs systematically better than its simple single-frame Gaussian mixture model (GMM) counterpart (99.1% vs. 92.2% in accuracy). Actually, the proposed sequential classifier wins over all its tested single-frame competitors (the best single-frame classifier is the nearest mean (NM) classifier which achieves up to 98.5% in accuracy). This highlights the relevance of exploiting the statistical correlation from human dynamics.

Continuous sensing and evaluating CPU-intensive prediction methods rapidly deplete a mobile system’s energy. Therefore, the third point requires a system to be energy-efficient enough for mobile-pervasive technologies. Several approaches have been introduced for this problem. Some methods aim to keep the number of necessary sensors low by adaptive selection [34] or based on the activity performed [35,36,37], for accurate activity prediction. Other approaches aim to reduce the computational cost by feature selection [38], feature learning [39], or proposing computationally inexpensive prediction models such as C4.5, random forest [40], or decision trees [41].

In this article, our methodology and experimental results are purely computational. Building and testing a prototype of such robotic, prosthetic legs for patient suffering from double trans-femoral amputation is the subject of our ongoing research. We expect that the patients may feel discomfort at the beginning but will become acclimated after a short adjustment period. We hope the prosthesis will be a useful tool in combating disability discrimination as is called for under several human rights treaties, such as the Rights of Persons with Disabilities convention by the United Nations [42] and Equality Acts [43,44] in jurisdictions worldwide. These also mandate access to goods, services, education, transportation, and employment. We expect the GaIn tool will be effective in helping patients tackle common obstacles such as stairs, uncut curb in urban areas.

The rest of the article is organized as follows. The next section gives a detailed description of the GaIn system. Section 3 describes the data collection and the performance evaluation methods used in our study. It also describes the feature extraction steps from the data obtained from the EMG and the motion sensors. In Section 4, we present our experimental results and discuss our findings. Finally, we conclude our study in the last section.

## 2. GaIn System

### 2.1. An Overview

The GaIn control system consists of two major parts. First, the controller recognizes the current activity of the user, and, second, it performs the necessary gain inference.

For the activity recognition, the GaIn system relies on a pair of triaxial accelerometer and gyroscope sensors installed symmetrically over the rectus femoris muscle (on thighs) 5 cm above the knee on the right and left legs, and on a pair of EMG sensors over the vastus lateralis (on both thighs) connected to the skin by three electrodes. For sensor locations, see Figure 2. The data from the accelerometers and gyroscopes are converted to angles and angular speed using the method described by Pedley [45]. Depending on the current recognized activity, the GaIn controller can perform the following actions:When the user is sitting, the controller does not allow any gait inference and the legs remain motionless. If an adequate amount of electrical activity from both thigh muscles is recognized by the EMG sensors on both thighs, then the system performs a standing up procedure.When the user is standing, then the controller can (i) keep the user in a standing position, (ii) start gait inference if one leg starts swinging, or (iii) perform a sitting down procedure if the electrical activity of both thigh muscles is suddenly high and both thighs have a similar position.When the user is walking, running, or taking stairs, then the controller performs gait inference using a recurrent neural network or goes to a standing position.

Figure 3 shows the possible transitions between different activities. For instance, if the user is walking, then the system cannot perform a sitting down activity without first stopping and standing, while, if the user is sitting, then the GaIn cannot infer walking-related activities without first standing up and standing.

### 2.2. Activity Recognition Method

For activity recognition, we used the RapidHARe model [26], which is a computationally inexpensive method for providing a smooth and accurate activity prediction with low prediction latency. It is based on a dynamic Bayesian network [46], illustrated in Figure 4, and the most likely activity $s_{t}$ being performed at time *t* with respect to a given data observed in a context window $v_{t},v_{t-1},\cdots ,v_{t-K}$ of length *K* is formulated by:(1)$${\hat{s}}_{t}={\mathrm{argmax}}_{s_{t}}\left\{\prod_{k=0}^{K}P\left(v_{t-k}|s_{t}\right)\right\},$$
where $P(v_{t-k}|s_{t})$ denotes the probability of the activity $s_{t}$ w.r.t a given observed data $v_{t-k}$. Conditional probabilities P(.∣.) are modeled with Gaussian mixture model (GMM) and the parameters were trained using the Expectation–Maximization (EM) method [47]. The training of GMMs was straightforward because our training data were segmented. For the full derivation of the model, we refer the reader to our previous work [26].

### 2.3. Gait Inference Method

The shank movement prediction was modeled with recurrent neural networks (RNNs) [48] with long-short term memory (LSTM) units [49]. Figure 5 shows the typical structure on an RNN. RNNs are universal mathematical tools for modeling statistical relationships in sequential data. While standard RNN cells are prone to the so-called forgetting phenomenon, LSTM cells aim to circumvent this shortcoming, as we describe below.

LSTM cells use two types of memory units to represent the past information of sequential data: one to capture short-term dependencies denoted by *h* and the other to capture long-term dependencies called state *c*. State *c* runs through the whole time and an LSTM performs four steps to update its data using so-called gates. The gates are: input gate, forget gate, input modulation, and output gate. The structure of an LSTM cell is shown in Figure 6. One of the main advantage of LSTMs is that each gate is differentiable, so their operations can be learned from data. The gates and the data manipulation steps are defined as follows:The forget gate calculates which information should be removed from state $c_{t}$ based on the hidden unit $h_{t-1}$ and the current input $x_{t}$. It is defined formally as $f_{t}=\sigma \left(W_{f}\left[v_{t},h_{t-1}\right]+b_{f}\right)$, where σ denotes the sigmoid function. The output $f_{t}$ can be considered as a bit vector, which indicates the components of the state vector *c* to be forgotten. For instance, $f_{t_{i}}\approx 1$ indicates that the value of the *i*th component of $f_{t}$ will be kept and $f_{t_{i}}\approx 0$ indicates that the value of that component will be forgotten.The input gate controls which information from the input should be kept and stored in the state vector $c_{t}$ at time step *t*. It is formally defined as $i_{t}=\sigma \left(W_{i}\left[v_{t},h_{t-1}\right]+b_{i}\right)$ and can be interpreted as a binary mask vector.The input modulation gate calculates a new candidate state vector ${\tilde{c}}_{t}=tanh\left(W_{g}\left[v_{t},h_{t-1}\right]+b_{g}\right)$.The new state vector $c_{t}$ is then calculated by $c_{t}=f_{t}\cdot c_{t-1}+i_{t}\cdot {\tilde{c}}_{t}$.The output gate decides which parts of the cell state go to the output. It is calculated by $o_{t}=\sigma \left(W_{o}\left[v_{t},h_{t-1}\right]+b_{o}\right)$.The new hidden state $h_{t}$ is formed from the new cell state whose values are first pushed between −1 and 1 using the tanh function, and then multiplied by the values of the output gate. Formally, $h_{t}=o_{t}\cdot tanh\left(c_{t}\right)$.Finally, the emission or the output of the cell (i.e., in our case, the predictions for the position of the shank) is calculated using $y_{t}=tanh\left(W_{y}h_{t}+b_{y}\right)$.

In the above, σ denotes the sigmoid function, ‘·’ represents element-wise or Hadamard product of vectors, [.,.] denotes vector concatenation, *W*s denote weight matrices, and *b*s denote the corresponding biases whose values are to be learnt from data.

In our work, the observed data $v_{t}$ is a four-component vector, in which each component corresponds to the angle and the angular speed of the left and right thighs, respectively. The angular data were calculated from two triaxial gyroscopes and accelerometer sensors located on the right and left thighs using the methods described by Pedley [45]. The RNN was trained to predict the angles of both shanks.

We do not recommend bidirectional RNNs or Hidden Markov models (HMMs) for gait inference. These methods require the whole observed sequence before making any predictions for intermediate time frames. In other words, bidirectional methods use data from the future to make a prediction in the present. This would increase the prediction latency [26].

The feet angle and position was not the subject of prediction because novel feet prostheses have good mechanical systems for feet positioning without any information [50].

## 3. Methods and Data Sets

### 3.1. Data Sets

In our experiments, we used the human gait data from the HuGaDB database [51]. The data is freely available at https://github.com/romanchereshnev/HuGaDB. This dataset consists of a total of five hours of data from 18 participants performing eight different activities. These participants were healthy young adults: four females and 14 males with an average age of 23.67 years (standard deviation [STD]: 3.69), an average height of 179.06 cm (STD: 9.85), and an average weight of 73.44 kg (STD: 16.67). The participants performed a combination of activities at normal speed in a casual way, and there were no obstacles placed in their way. For instance, starting in the sitting position, participants were instructed to perform the following activities: sitting, standing up, walking, going up the stairs, walking, and sitting down. The experimenter recorded the data continually using a laptop and annotated the data with the activities performed. This provided us a long, continuous sequence of segmented data annotated with activities. In total, 1,077,221 samples were collected. Table 1 summarizes the activities recorded and provides other characteristics of the data.

During data collection, MPU9250 inertial sensors and electromyography sensors made in the Laboratory of Applied Cybernetics Systems, Moscow Institute of Physics and Technology (www.mipt.ru) were used. Each EMG sensor has a voltage gain of about 5000 and a band-pass filter with bandwidth corresponding to a power spectrum of EMG (10–500 Hz). The sample rate of each EMG-channel is 1.0 kHz, the analog-to-digital converter (ADC) resolution is 8 bits, and the input voltages is 0–5 V. The inertial sensors consisted of a three-axis accelerometer and a three-axis gyroscope integrated into a single chip. Data were collected with the accelerometer’s range equal to ±2 g with sensitivity 16.384 least significant bits (LSB)/g and the gyroscope’s range equal to $\pm 2000^{{}^{\circ}}$/s with sensitivity 16.4 LSB/°/s. All sensors were powered with a battery, which helped to minimize electrical grid noise.

Accelerometer and gyroscope signals were stored in int16 format. EMG signals were stored in uint8. In our experiments, all data were scaled to the range [−1,1].

In total, six pieces of inertial sensors (three-axis accelerometer and three-axis gyroscope) and two EMG sensors were installed symmetrically on the right and left legs with elastic bands. A pair of inertial sensors were installed over the rectus femoris muscle 5 cm above the knee, another pair of sensors around the middle of the shin bone at the level where the calf muscle ends, and a third pair on the feet on the metatarsal bones. Two EMG sensors were connected to the skin by three electrodes over the vastus lateralis muscle. The EMG sensors additionally provided two more features. This installation provided 36 signal sources, 18 signal sources were from the accelerometers (2 legs × 3 sensors × 3 axes), 18 signals were from the gyroscopes (2 legs × 3 sensors × 3 axes), and two signals were from the EMG (2 sensors). For the sensor locations, we refer the reader to see [51].

The sensors were connected through wires with each other and to a microcontroller box, which contained an Arduino electronics platform with a Bluetooth module. The microcontroller collected 56,350 samples per second on average, with a standard deviation of 3.2057, and then transmitted them to a laptop through a Bluetooth connection. Data acquisition was carried out mainly inside a building. Data were not recorded on a treadmill. We note that walking contains turning but unfortunately the annotation does not indicate this information.

We note that some data in HuGaDB contained corrupted signals and, typically, several gyroscope measurements were overflown and hence trimmed. We discarded these data from our experiments and Table 1 summarizes information on data we actually used.

### 3.2. Feature Extraction Methods

First, raw data obtained from the gyroscope and accelerometer sensors were filtered with moving average using a window of 100 samples. This was performed to remove the bias drift of inertial sensors [52].

The gait inference method is based on the thigh angle and angular speed data in the sagittal plane. The initial angle degrees for thigh and shank are calculated based on the accelerometer data and Earth gravity [45]. Formally, the start angle of the thigh (θ) is calculated with
(2)$$\theta_{start}=\arctan\left(\frac{T_{y}}{\sqrt{T_{x}^{2}+T_{z}^{2}}}\right),$$
where $T_{x},T_{y},$ and $T_{z}$ denote the values of the accelerometer sensors located on thigh. Similarly, the start angle of the shank (ϕ) is calculated via
(3)$$\phi_{start}=\arctan\left(\frac{S_{y}}{\sqrt{S_{x}^{2}+S_{z}^{2}}}\right),$$
where $S_{x},S_{y}$, and $S_{z}$ denote the values of the accelerometer sensors located on shank. The angular velocities are from the gyroscope data. Let $\omega_{T}\left(t\right)$ and $\omega_{S}\left(t\right)$ be angular velocities of thigh and shank at time *t*, respectively. The angles of thigh θ(t) and shank ϕ(t) at time *t* can be calculated as follows [45]:(4)$$\begin{array}{ccc} \theta (t) & = & \theta_{start}+\int \omega_{T}\left(t\right)dt, \end{array}$$(5)$$\begin{array}{ccc} \phi (t) & = & \phi_{start}+\int \omega_{S}\left(t\right)dt. \end{array}$$

For every time frame *t*, the standard deviation (std) of the gradients of the EMG signals was calculated from the previous 5 and 10 measurements and were used for sitting down and standing up intention recognition, respectively. Formally, let ϵ(t) denote the EMG signal data at time *t*. The feature for the variance of the gradients of the EMG signals in the last *w* time steps is formulated as $v_{w}\left(t\right)={\mathrm{STD}}_{i=0\cdots w}\left\{\partial \epsilon \left(t-i\right)\right\}$. For sitting down intention recognition, we use a feature to measure the variance of the differences of the accelerometer data between the two thighs in the last 10 data frames. It is formulated as $d_{x}\left(t\right)={\mathrm{STD}}_{i=0\cdots 10}\left\{\right|T_{x}^{L}\left(t-i\right)-T_{x}^{R}\left(t-i\right)\left|\right\}$ for the *x*-axis.

### 3.3. Model Implementation Details

The GaIn system (a) recognizes activities and intentions and (b) infers gait. We used a RapidHARe module to recognize standing up intention in the sitting position from the EMG sensor data. The intention was modeled with 10 Gaussian components, while sitting was modeled with one Gaussian component. We used another RapidHARe model to recognize sitting down intention during standing or walking activities from EMG sensor data and the differences of the accelerometer data. The intention was modeled with 5, while all others were modeled with two Gaussian components, respectively. We used a third RapidHARe module to recognize sitting, standing, and walking-related activities using one Gaussian component for each. All models used 20 long context windows.

For gait inference, the RNN consisted of 50 LSTM hidden units in one hidden layer. The learning objective for the RNN was to minimize the squared error between the predicted and the true shank angle. For the training, the input sequential data were chunked into 15 long data segments. Table 2 summarizes the features we used in each module.

The classification methods were implemented using the Python scikit-learn package (version 0.18.1). The RNN/LSTM was implemented with Keras library (version 2.1.2) using TensorFlow framework as backend (version 1.4.0). Training and testing were carried out on a PC equipped with Intel Core i7-4790 CPU, 8 Gb DDR-III 2400 MHz RAM, and Nvidia GTX Titan X GPU. Training and testing on all data took roughly three hours.

### 3.4. Evaluation Methods

The performance of our GaIn method was evaluated using a supervised cross-validation approach [53]. In this approach, data from a designated participant were held out for tests, and the rest of the data from the 17 participants were used for training. Thus, this approach gives a reliable estimation of how well the GaIn system would perform for a new patient whose data have not been seen before. In our experiments, we repeated this test for every user in the dataset and averaged the results.

The error of the gait inference was measured by the absolute value of the difference between the true and the predicted shank angles. The activity recognition was evaluated with $Precision=\frac{TP}{TP+FP}$ and $Recall=\frac{TP}{TP+FN}$ metrics, where TP, FP, and FN denote the number of the true positive, false positive, and false negative predictions, respectively. In addition, we calculated and reported the $F_{1}$ score, which is a combined score of the recall and the precision measures, defined as $F_{1}=2\frac{Precision\cdot Recall}{Precision+Recall}$.

## 4. Results and Discussion

Our overall results on the gait inference can be seen in a video at https://youtu.be/aTeYPGxncnA, while two screenshots are shown in Figure 7. Many videos have been generated on different data recorded on different participants, but we did not see visually notable differences in the videos. The reason we chose this example particularly is that this data contains a variety of activities during a relatively short time.

Figure 8 shows the inference for continuous series of standing up, sitting down, and a few walking-related activities. Note that the standing up and the sitting down activities are inferred based on the variance of the gradients in the EMG signals obtained from both thigh muscles (shown with green lines), and that the shank degrees (shown with black lines) are irrelevant here. The lower part of the figure indicates the true activities performed by the participant, while the upper part indicates the recognized activities. We note that the length of the sitting down and standing up activities in the figures is irrelevant here because the length would depend on how the robotic prosthetic legs performed these movements once the patient’s intention was recognized. The shank movement inference during walking-related activities is based on the thigh angles (not shown), and the EMG sensor data is ignored here. To help guide the reader, we have also indicated the current walking type by the color of the background, but GaIn does not take this information into account.

### 4.1. Activity Classification Results

First, we discuss our experiments on how efficiently the activity recognition module of the GaIn system recognizes the patient’s intention to (a) sit down from a standing position and (b) to stand up from a sitting position using mainly EMG signals. The results, summarized in Table 3, show that standing and sitting position recognition can be achieved with high accuracy; however, it is easier to recognize standing up intention than sitting down intention. Our system achieved 0.99 recall and 0.99 precision for recognizing standing up intention, but it achieved only 0.68 recall and 0.99 precision for sitting down activity. The reason is that the muscle activity in both thighs is very low in a sitting position, thus it is effortless to recognize standing up intention form the sudden increase in muscle activity. However, muscle activity is already present in a standing position, which makes it more challenging to distinguish a patient’s simple balancing or walking efforts from a sitting down intention. Nevertheless, incorrect activity prediction can result in different impacts on the patient. When the GaIn system incorrectly recognized a standing up activity while the user is sitting, then the system simply stretched the robotic prosthetic leg, resulting in no harm to the patient. However, when a sitting down intention is predicted while the user is simply standing, then the patient would fall and may suffer serious injury. In our opinion, it is more important to achieve lower false alarm (high precision) than missed alarm (high recall) rates for sitting down activity. Therefore, we calibrated the decision threshold so that the activity recognition module achieved as high as 0.99 precision at the expense of recall, which decreased to 0.68. As a consequence, users may need to produce clearer and longer signals to the system for sitting down, but this results in GaIn causing fewer injuries from falling. Figure 9 shows the system in action with different users having different qualities of EMG signals.

We examined the prediction latency and plotted a histogram of the activity recognition lag time in Figure 10. On average, it takes 602 milliseconds to recognize standing up intention (shown in Figure 10A), while it takes 846 milliseconds to recognize sitting down (shown in Figure 10B). We note that the standing up recognition has lower variance than the sitting down recognition and this is concluded from the width of the distributions in Figure 10. Note that the higher lag time for sitting down recognition is a result of the threshold calibration, as discussed above.

Finally, we mention that the quality of the EMG signals greatly depends on the physical properties of the user’s skin. Some users generated poor EMG signals (see the results for participant ID = 5, 14 in Table 3) that hampered the activity recognition consistently, while some users generated good quality EMG signals (see the results for participant ID = 1, 6 in Table 3), resulting in almost perfect activity recognition. Therefore, to mitigate dependency on the EMG signals, we propose calibrating the system’s activity recognition module for each patient individually in the future.

### 4.2. Gait Inference Results

The results for gait inference are shown in Figure 11 for various walking-related activities such as walking, running, and taking the stairs up and down. The dashed lines show the true angle of the shank, while the solid line shows the prediction for the shank angle. The line segments going upward correspond to swing phases and line segments going downward correspond to stance phases in the gait cycle. The error, the difference between the true and the predicted movements, is indicated by the shaded area. It appears that the errors occur at the peaks and troughs which correspond to the turning point between the swing and stance phases. The color of the background indicates activity performed. Note that these activity labels were not incorporated into the training procedure; they are presented simply for illustration purposes. The prediction errors for different activities are listed in Table 4. The error was calculated for each activity over all data of all users. The average of the prediction errors for the shank angles across different activities is 4.55 degrees. Figure 12 shows the coordination and coordination variability of the true (red) and predicted (blue) shank angles with respect to the thigh angles. Here, we used the same data as for Figure 1. This scatter plot shows that the predicted shank angles are in accordance with the true shank angles. However, the predicted shank angles do not span over the range of the true angles in some cases. For instance, in plot F, the predicted shank angles do not reach the extremes of the true angles. This is the error that occurs at peaks and troughs in Figure 11.

### 4.3. Variance in Different Phases

People walk differently, resulting in variance in gaits [1]. Moreover, gait varies over different cycles for the same person as well. Figure 1 shows this natural variance. This variance prevents achieving 100% accuracy in gait prediction for someone’s gait based on other people’s gait data. It has also been noticed that variance in the swing phase is larger than in the stance phase [54]. This is as expected, since the stance phase is more important in walking stability, while legs may move more freely in the swing phase [54]. We also observed this fact in our data and plotted the shank angles in the stance and swing phases of one gait cycle obtained from different users. In Figure 13, panel A shows the shank angles of the gait cycle in the stance phase (blue lines) and the variance (red line) and panel B shows the same information for the swing phase. The figure shows that the variance is higher in the swing phase. Therefore, we expect higher prediction errors for the swing phase than for the stance phase. In fact, the mean shank degree prediction error across all activities is 4.783 (STD: 1.171) in the stance phase and 6.182 (STD: 1.680) in the swing phase. Table 5 shows detailed prediction errors for different activities.

### 4.4. Inference Errors around Activity Change

We closely examined the errors around activity changes; for instance, when a walking user started running. We measured the gait inference errors in a range of ±15 data samples (equivalent to half of a second) around the true activity change. We found that the shank degree prediction error is 5.44°, which is not especially larger than general. The detailed results for different activity transitions are shown in Table 6.

## 5. Conclusions

In this article, we presented a new method, called GaIn, for human gait inference. GaIn was designed to predict the movements of the lower legs based on the movements of both thighs. This can potentially be the basis for building non-invasive, robotic, lower limbic prostheses for patients suffering from double trans-femoral amputation. Our method is based on the observation that the thigh degrees strongly correlate to the shin bone degrees during casual walking-related activities. In this article, we showed that the shank degrees can be predicted using recurrent neural networks with LSTM memory cells using thigh degrees as input. Our experimental results showed that our system is highly accurate and it achieved 4.55 degree prediction error on average. The error for the stance phase was even lower. We believe that a recurrent neural network is a suitable mathematical model to simulate the motor cortex of the human neural system, and we think this is the reason why GaIn achieves low prediction error.

However, in a real-life application, sitting down and standing up intentions cannot be recognized from thigh movements. To circumvent this, we applied EMG sensors placed on the skin over the vastus lateralis—the thigh muscles; therefore, the patient can signal her/his intentions by increasing thigh muscle activity. Our system achieved a 99% precision and recall in recognizing standing up intention, and achieved 99% precision and 68% recall in recognizing sitting down intention. We mentioned that a patient may suffer injury if the system incorrectly predicts a sitting down intention during walking or just standing. For safety reasons, we adjusted the decision rule accordingly to maintain low false alarm (high precision) at the expense of high missed alarm (low recall). As a result, users may need to produce clearer signals to indicate sitting down intention.

Here, we presented our results purely on in silico experiments. Building a real prototype of such a robotic prosthetic leg is the subject of our current research. In practice, we expect that the patients may feel a little discomfort using such robotic prosthetic legs at the beginning and need to adjust to the device and, on the other hand, we will also need to make adjustment in the GaIn model to adapt the prosthesis to diverse peoples and urban situations. We hope that the prosthesis will be a useful tool in combating disability discrimination.

## Figures and Tables

**Figure 1 sensors-18-04146-f001:**
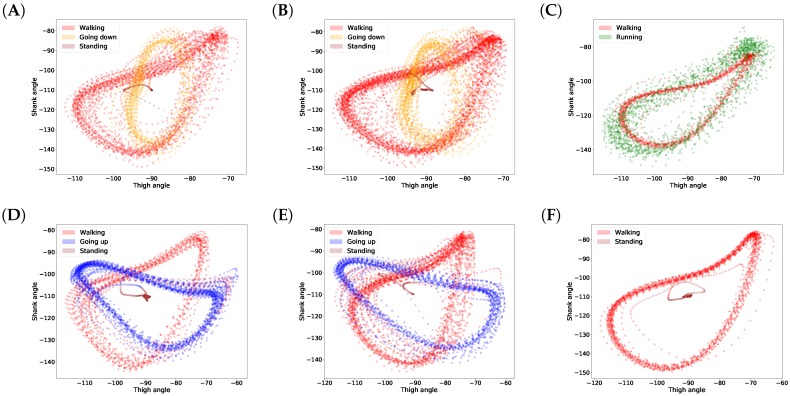
Correlations between shank and thigh movement over several gait cycles in different activities. The angles of the thigh and shank are measured to horizontal line (see Figure 2). Plots (**A**–**F**) show various examples.

**Figure 2 sensors-18-04146-f002:**
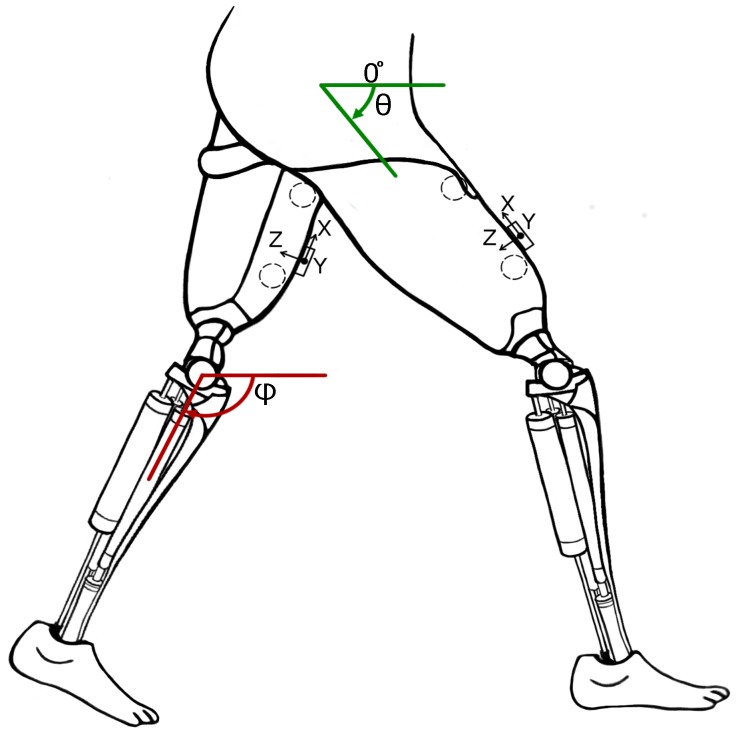
Concept of robotic prosthetic legs for patients suffering from double trans-femoral amputation. Circles show the location of EMG sensors and boxes show the location of accelerometers and gyroscopes.

**Figure 3 sensors-18-04146-f003:**
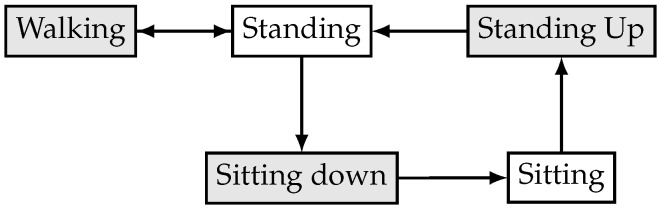
Activity transition graph of the GaIn controlling system.

**Figure 4 sensors-18-04146-f004:**
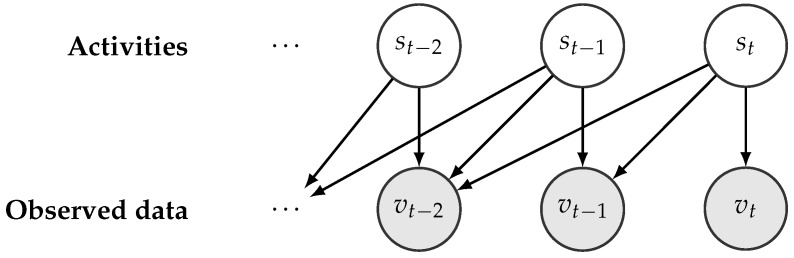
Illustration of an unfolded dynamic Bayesian network w.r.t a given activity series.

**Figure 5 sensors-18-04146-f005:**
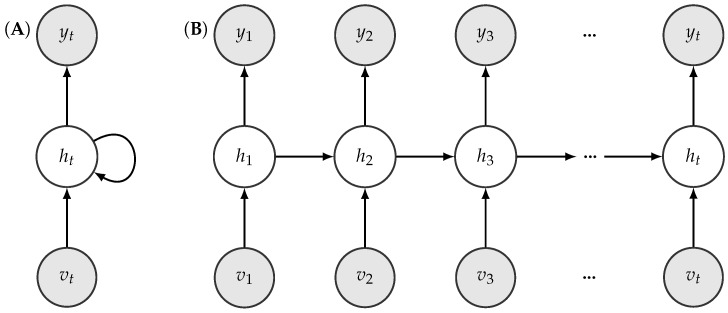
Illustration of a folded (**A**) and an unfolded (**B**) recurrent neural network structure.

**Figure 6 sensors-18-04146-f006:**
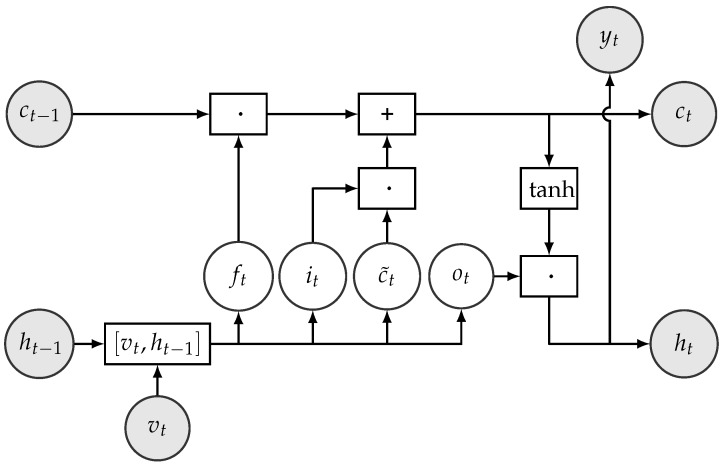
Structure of LSTM cell. The operators ·, +, [.,.], and tanh in boxes represent element-wise multiplication, addition, concatenation, and tanh operations on vectors, respectively.

**Figure 7 sensors-18-04146-f007:**
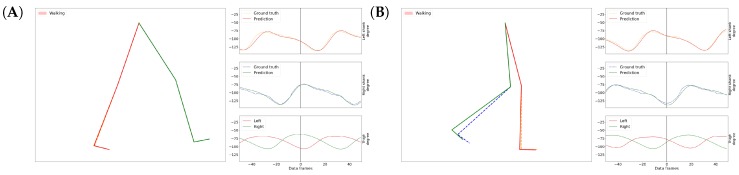
Screenshots of GaIn during gait inference (**A**,**B**). Around 56 data frames add up to one second. See the full video at: https://youtu.be/aTeYPGxncnA.

**Figure 8 sensors-18-04146-f008:**
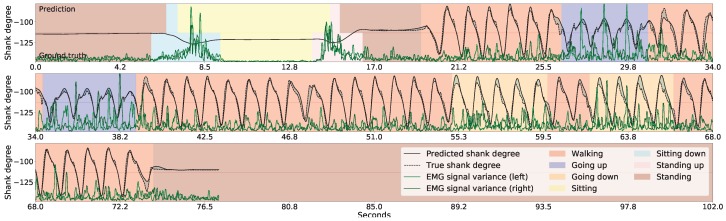
Gait inference and activity recognition using GaIn. The first part (from 4.2 s to 20 s) shows sitting-related predictions, while the second part (from 21 s) shows shank movement inference for walking related activities. During the first part, the sitting down and standing up intentions are recognized based on the EMG signal variance (green) while the shank angles (black) are irrelevant here—while, in the second part, the shank movement inference for walking is calculated from the thigh angles (not shown) and the EMG signals are disregarded here. The walking type is indicated by the colors in the background only for the reader; it was not used in our methods.

**Figure 9 sensors-18-04146-f009:**
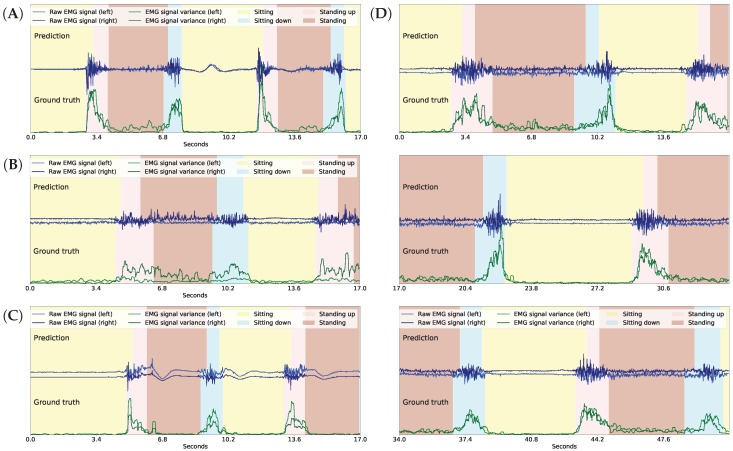
Activity recognition in GaIn with good (**A**), “waving” (**B**), weak (**C**), and “waving” and weak (**D**) EMG signals from participants ID = 1,7,12,16, respectively. Note that plot D consists of three continuing panels.

**Figure 10 sensors-18-04146-f010:**
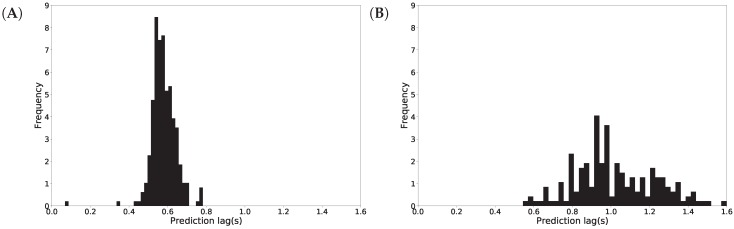
Activity recognition latency in seconds (s) for standing up (**A**) and sitting down (**B**).

**Figure 11 sensors-18-04146-f011:**
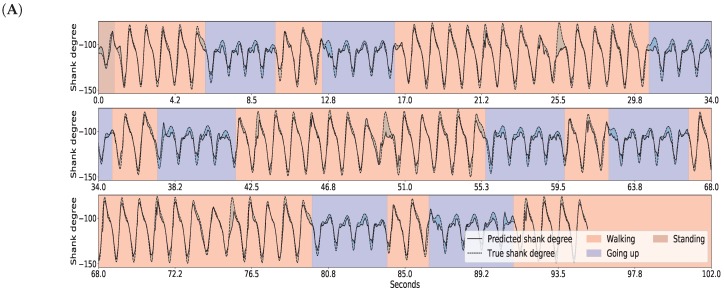
GaIn inference for various walking-related activities. The activity is indicated by the background color for the reader, but this information was not used in the methods. The shank degree is predicted based on thigh angles (not shown). The solid black line shows the predicted, the dashed line shows the true angles of the right shank, while the shaded area between them indicates the prediction error. Plots for the left leg are similar. Plots (**A**–**C**) show various examples.

**Figure 12 sensors-18-04146-f012:**
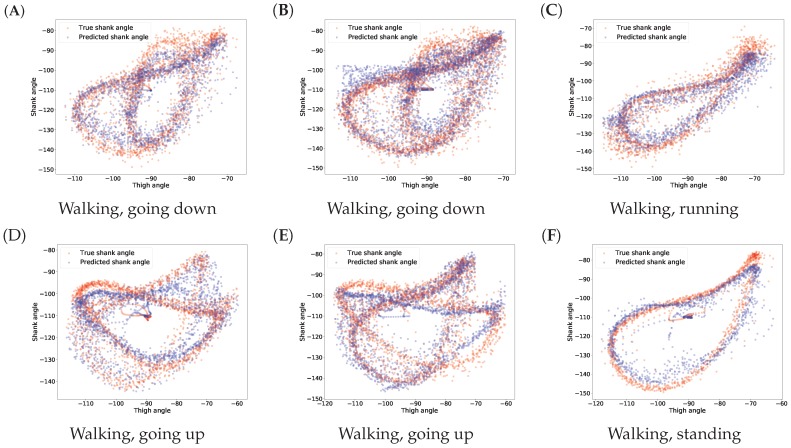
Predicted and true shank angles as a function of thigh position over several gait cycles in different activities. The input data used was the same as in Figure 1. The caption below each figure indicates the types of activities performed on the plot. Plots (**A**–**F**) show various examples.

**Figure 13 sensors-18-04146-f013:**
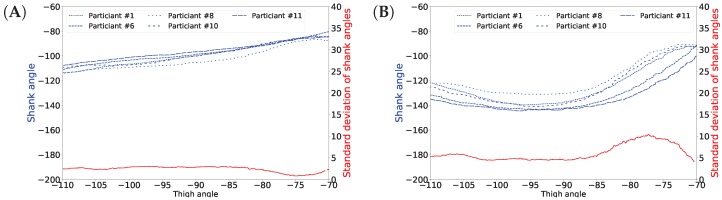
Shank angles of different participants in stance phase (**A**) and swing phase (**B**).

**Table 1 sensors-18-04146-t001:** Characteristics of data and activities.

Activity	Time Sec (min)	Percent	Samples	Description
Walking	5354 (89)	29.24	314,949	Walking and turning at various speeds on a flat surface
Running	1120 (18)	6.12	65,917	Running at various paces
Going up	2297 (38)	12.54	135,127	Taking stairs up at various speeds
Going down	2036 (33)	11.12	119,819	Taking the stairs down at various speeds and steps
Sitting	1407 (23)	7.69	82,819	Sitting on chair; floor not included
Sitting down	309 (5)	1.69	18,180	Sitting on chair; floor not included
Standing up	350 (5)	1.91	20,603	Standing up from a chair
Standing	5436 (90)	29.69	319,807	Static standing on a solid surface
Total	18,312 (305)	100.0	1,077,221	

**Table 2 sensors-18-04146-t002:** Features used in GaIN. All features were scaled between 0 and 1.

Model	Features	Window Length
Gait inference	Left thigh angle $\theta^{L}\left(t\right)$,	15
	Right thigh angle $\theta^{R}\left(t\right)$,	
	Left thigh angular speed $\omega_{T}^{L}\left(t\right)$,	
	Right thigh angular speed $\omega_{T}^{R}\left(t\right)$.	
Standing up	EMG sensor data variance from the left thigh $v_{5}^{L}\left(t\right)$,	20
	EMG sensor data variance from the right thigh $v_{5}^{R}\left(t\right)$.	
Sitting down	EMG sensor data variance from the left thigh $v_{10}^{L}\left(t\right)$,	20
	EMG sensor data variance from the right thigh $v_{10}^{R}\left(t\right)$,	
	Variance of the differences on *x*-axis accelerometer data $d_{x}\left(t\right)$,	
	Variance of the differences on *z*-axis accelerometer data $d_{z}\left(t\right)$.	
Sitting-Standing	*x*-axis accelerometer data from left thigh: $T_{x}^{L}$,	20
	*x*-axis accelerometer data from right thigh: $T_{x}^{R}$,	
	*z*-axis accelerometer data from left thigh: $T_{z}^{L}$,	
	*z*-axis accelerometer data from right thigh: $T_{z}^{R}$.	

**Table 3 sensors-18-04146-t003:** Classification results for each participant.

Participant ID	1	2	3	4	5	6	7	8	9	10	11	12	13	14	15	16	17	18	Mean
standing up recall	1	1	1	1	1	1	1	1	1	1	1	1	1	1	1	0.93	1	1	0.99
standing up precision	1	1	0.94	1	1	1	1	1	0.94	1	1	1	1	1	0.94	0.93	1	1	0.99
standing up $F_{1}$	1	1	0.97	1	1	1	1	1	0.97	1	1	1	1	1	0.97	0.93	1	1	0.99
sitting down recall	1	0.78	1	0.50	0.19	1	0.50	0.82	0.93	0.86	0.5	0.88	0.56	0.38	0.60	0.64	0.70	0.88	0.68
sitting down precision	1	1	0.90	1	1	1	1	1	0.93	1	1	1	1	1	1	0.90	1	1	0.99
sitting down $F_{1}$	1	0.88	0.95	0.67	0.32	1	0.67	0.90	0.93	0.92	0.11	0.93	0.71	0.55	0.75	0.75	0.82	0.93	0.77
standing recall	1	1	1	1	1	1	1	1	1	0.96	1	0.97	1	1	1	1	1	1	0.99
standing precision	1	1	1	0.94	1	1	1	1	1	1	1	1	1	1	1	1	1	1	0.99
standing $F_{1}$	1	1	1	0.97	1	1	1	1	1	0.98	1	0.98	1	1	1	1	1	1	0.99
sitting recall	1	1	1	1	1	1	0.95	1	1	1	1	1	1	1	1	1	1	1	0.99
sitting precision	1	0.78	1	1	1	1	1	1	1	1	1	1	1	1	1	1	1	1	0.99
sitting $F_{1}$	1	0.88	1	1	1	1	0.98	1	1	1	1	1	1	1	1	1	1	1	0.99

Note that some participants (e.g., ID = 5, 14) yield poor results due to weak EMG sensor signals. These cases could be circumvented using individual EMG signal calibration.

**Table 4 sensors-18-04146-t004:** Gait inference error.

	Walking	Running	Going up	Going down	Standing	Mean
Mean	4.988	5.648	5.820	5.148	1.174	4.555
Std	0.910	2.212	1.299	1.158	0.457	1.207

Error measured in absolute difference between the true and the predicted shank angles in degrees.

**Table 5 sensors-18-04146-t005:** GaIn inference error.

	Walking	Running	Going up	Going down	Mean
*Swing phase*					
Mean	5.826	6.420	6.738	5.744	6.182
Std	1.0817	2.750	1.437	1.452	1.680
*Stance phase*					
Mean	4.268	4.967	5.140	4.758	4.783
Std	0.800	1.700	1.215	0.969	1.171

Error measured in degrees.

**Table 6 sensors-18-04146-t006:** Average shank degree prediction error at activity transitions.

Activity Transition	Mean	Std
walking → running	5.79	2.297
walking → going up	5.34	1.417
walking → going down	5.68	0.959
walking → standing	4.50	0.742
running → walking	5.31	2.352
going up → walking	5.15	1.661
going up → standing	7.24	0.837
going down → going up	6.21	0.479
going down → walking	6.22	1.734
standing → going up	5.75	1.331
standing → walking	4.20	3.065
standing → going down	6.11	2.242
Mean	5.44	1.471

The degree error was measured in ±15 sample interval (around half a second long range) at the true activity transition border.

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
