# Peer review of "GaIn: Human Gait Inference for Lower Limbic Prostheses for Patients Suffering from Double Trans-Femoral Amputation"

_sensors, 2018, doi:10.3390/s18124146_

Reviewer 1 Report

This article uses a machine learning approach to infer the movement of the shank and predict intention for different types of walking activities based on inertial movements and electromyography of the thigh.  The results of this study may have implications for developing robotic prosthetics for double trans-femoral amputees.  Overall, the manuscript reads well with plenty of information.

My biggest concern is putting the results of this study in a real-world context.  For example, the error in predicting the shank angle is provided and it is stated that this error is low.  However, what is considered low, and how was this determined?  Can you reference real-world shank angle variance?  Is there a clinically important difference?  How does error in shank angle influence joint angles, which is a more commonly-reported variable?  In what plane or dimension is the shank angle reported?  Also, the absolute value of the shank angle error was reported, but does the direction matter?  From Figure 11, it appears that the errors occur at the peaks and troughs, so that may be justification for reporting the absolute value.  What does this mean for real-world implications?  At what point in the gait cycle do peak in shank angle occur?  Can you relate this to Figure 1?  Presumably, the same plot for Figure 1 created with predicted shank angle data would be more squished as the predicted shank angles have a smaller range.  Also, the type of plot in Figure 1 is often used to define coordination and coordination variability.  How would the predicted shank angles influence coordination and coordination variability?

Introduction
Line 25-26: Provide more information about what constitutes “good gait” and “unusual gait”.
Line 48: What is meant by “at most”?
Line 50: I don’t think the parentheses are needed.
Line 73: Should be “gait” or “gait patterns” (not “gaits”).
Line 82-87: These sentences are a bit awkward and should be revised for clarity.  In particular, the phrases “tackle down” and “uncut curb” are not familiar.

Methods
Throughout the paper, it has been indicated that the sensors (IMU and EMG) were placed “on” muscles.  However, they were likely placed on the skin above the muscles.  I would suggest rewording these instances to include the words “above” or “recorded from”.  (e.g. Line 98-99, 199, 336-337)
Line 197-203: This description could be clarified a bit.  Does “pair” mean “bilateral”?  What is meant by “six pieces”?  Were the accelerometer and gyroscope separate?  The number and type of features could also be better clarified – perhaps by explicitly stating that 18 features were from the accelerometer (6 sensors x 3 axes), 18 features were from the gyroscope (6 sensors x 3 axes), and 2 features were from the EMG (2 sensors).  Also, I’m not sure “features” is the right term here as I believe it is referring just to the raw signal, and the next section discusses feature extraction.

Results and Discussion
Line 249: “on” should be “in”
Figure 7 and Video: Thigh is spelled incorrectly on the y-axis.  Were these figures should for different people or the same person?  Was it a representation of the best or worst inference?  Specific to the video, the legend for the thigh figure moves around and is a bit distracting.
Figure 8: The scale for the EMG variance signal is small, and should be the main focus of the figure as that is what is used to predict the activity.  Likewise, if the shank degree is irrelevant, should it be removed from this figure?
Figure 9: As with the comment for Figure 8, the scale of the EMG variance could be increased to better show this variable.  Also, plots A, B, C and D are labeled and mentioned in the caption.  What are the other two unlabeled plots?
Line 283-284: What is “roughly low variance” and how was this determined?
Line 288: What is “poor EMG signals” and how was this determined?  Is there an example or definition of what makes it poor?
Line 301: Is the overall shank angle error the mean of each type or the mean over all data points?  The amount of data for each type of walking is different, so it is unclear if the error is a weighted average.
Figure 10: I would recommend putting both of these plots (A and B) on the same x-axis and y-axis scale to show the difference in variability.
Line 308: The statement that “stance phase is more important in walking stability” needs support – please provide a reference.  Also, consider that foot contact with the ground provides an additional constraint than no foot contact – perhaps this alone is the reason for reduced variability of the shank angle during stance compared to swing.
Line 314-315, and 320: Similar to the comment above, explain how the overall prediction error was determined (across all activities or across all data points).
Line 320: Please specify if the activity change points in this analysis were the actual or predicted activity change points.
Figure 11: When describing the different activities in each plot, specify whether it is plot A, B or C – this will probably require some rewording of the first sentence.  Also, indicate within the caption that the plots are for the right leg.  In the last sentence, “is” should be “are”.
Figure 12: “Standard” is spelled incorrectly on the y-axes (plots A and B).

Conclusion
Line 331-332: Consider revising to “… with lower error for the more important stance phase compared to swing phase.”  However, this sentence may need to be changed based on a previous comment.

Author Response

Please find our responses in the attached pdf file.

Author Response

(The authors gave the same response as above.)

Reviewer 3 Report

This study proposed a new method called GaIn to predict the movement of lower leg based on movement of the thigh with using accelerometer, gyroscope, and EMG sensor placed on Subject’s lower limb. Although the results on this study seems to be good, there is still some parts in the manuscript, especially on method section, need to be better explained, e.g. the part describing the training with LSTM.

Overall, this proposed method contained a good idea to help people who has bilateral lower limb amputation to be able to walk again; however, the safety issue related to sitting down and standing up intentions need to be addressed more seriously. Other concern is the usage of EMG signal being sensitive to inter-subject variability which need to be addressed by providing some calibration procedure and/or signal normalization process.

Some detail comments for this study are shown below:

1.       The author stated that “Several studies have analyzed human gait data obtained from inertial gyroscope and accelerometer sensors mounted on different parts of the body. In this article, we take a step further in gait analysis and provide a methodology for predicting the movements of the missing parts of the legs. ” To better clarify the key function of the methodology, I would suggest the authors to revise the sentence. For example, “Several studies have analyzed human gait data obtained from inertial gyroscope ….. a step further in gait analysis and provide a methodology for predicting the movements which can be further applied in prosthesis which act to replace the missing part of the leg during walking…”
2.       Line 2018-220, How did the authors process the EMG signal before being used as the input for the model?
How did the authors deal with EMG movement artifact which might happen during the trial?
3.       Line 249, Figure 7 needs bigger size and better resolution
4.       Line 287-292, The authors proposed calibrating the system to mitigate the dependency on EMG signal; however the proposed calibrating method did not clearly explain on the manuscript?
Did the authors try to normalize (e.g., time normalization relative to movement cycle, or amplitude normalization relative to signal obtained during maximal voluntary contraction) the EMG signal before input to the model?
5.       Considering the application of current method in real life condition, would it become problem if later the EMG system used in testing was different from the EMG system used on the training phase? Also how will the performance of the method possibly alter if the user applied on the same muscle but different positions?
My concern is in the application phase, when the subject need to use the EMG sensors, is the model robust enough to the variation of electrode placement (even in the same muscle) or the inter-subject variability in terms of EMG signal?
6.       Is the EMG sensor used on this study is commercially available? If not, how about its validity and reliability? If yes, please provide citations in which this sensor was validated
7.       Line 233, The authors said the method were implemented using Python scikit-learn package and the PC system included Titan X GPU. However, I am wondering whether the current Scikit-learn is capable to support training with GPU. I would suggest the authors to provide information in the statement to clarify which framework was used for training the RNN/LSTM model with GPU, or it was just trained with CPU? In addition, how long the training session takes with that PC’s specification is worth mentioned in the manuscript.

Author Response

Please find our responses in the attached pdf file.

Round  2

Reviewer 3 Report

Authors already satisfied all the Reviewer's comment and suggestion, therefore I recommend this article to be accepted.